# Comprehensive Proteome Profiling of a *Xanthomonas campestris* pv. Campestris B100 Culture Grown in Minimal Medium with a Specific Focus on Nutrient Consumption and Xanthan Biosynthesis

**DOI:** 10.3390/proteomes12020012

**Published:** 2024-04-03

**Authors:** Ben Struck, Sanne Jitske Wiersma, Vera Ortseifen, Alfred Pühler, Karsten Niehaus

**Affiliations:** 1Department of Biology, Bielefeld University, Universitätsstraße 25, D-33615 Bielefeld, Germanyswiersma@cebitec.uni-bielefeld.de (S.J.W.);; 2Center for Biotechnology (CeBiTec), Bielefeld University, Universitätsstraße 27, D-33615 Bielefeld, Germany; puehler@cebitec.uni-bielefeld.de

**Keywords:** xanthan, polysaccharide, *Xanthomonas campestris*, course of time, LC-MS/MS, semiquantitative, data-dependent acquisition

## Abstract

Xanthan, a bacterial polysaccharide, is widespread in industrial applications, particularly as a food additive. However, little is known about the process of xanthan synthesis on the proteome level, even though *Xanthomonas campestris* is frequently used for xanthan fermentation. A label-free LC-MS/MS method was employed to study the protein changes during xanthan fermentation in minimal medium. According to the reference database, 2416 proteins were identified, representing 54.75 % of the proteome. The study examined changes in protein abundances concerning the growth phase and xanthan productivity. Throughout the experiment, changes in nitrate concentration appeared to affect the abundance of most proteins involved in nitrogen metabolism, except Gdh and GlnA. Proteins involved in sugar nucleotide metabolism stay unchanged across all growth phases. Apart from GumD, GumB, and GumC, the gum proteins showed no significant changes throughout the experiment. GumD, the first enzyme in the assembly of the xanthan-repeating unit, peaked during the early stationary phase but decreased during the late stationary phase. GumB and GumC, which are involved in exporting xanthan, increased significantly during the stationary phase. This study suggests that a potential bottleneck for xanthan productivity does not reside in the abundance of proteins directly involved in the synthesis pathways.

## 1. Introduction

Microorganisms have developed coping strategies for physical, chemical, or biological stress factors [1]. One of these strategies is the production of extracellular polysaccharides (EPSs), which can protect bacteria, microalgae, fungi, and protists against desiccation [2], salt stress [3], and extreme temperatures [4,5,6]. EPSs have also become vital in various biotechnological applications, and xanthan gum is the most commercially important polysaccharide of microbial origin, with an expected market growth rate of 5.6% between 2020 and 2030 at a market size of USD 1.2 billion [7]. Xanthan is produced by the Gram-negative γ-proteobacteria genus Xanthomonas [8] and is a crucial constituent of their biofilms, important for their phytopathogenic lifestyle [9,10]. *Xanthomonas campestris* (*Xcc*) is being extensively researched due to its xanthan production [11].

Xanthan gum consists of a pentameric-repeating unit, per which two β1,4-linked and β1,2-linked D-glucose molecules form a cellulose-like backbone that contains a side chain of β1,4-linked D-mannose, D-glucuronic acid, and β1,2-linked D-mannose [12]. The gum gene cluster encodes the proteins for the synthesis of xanthan [13]. Xanthan synthesis starts with the transfer of glucose to a lipid carrier catalyzed by GumD, after which GumM, GumH, GumK, and GumI then add the remaining sugar moieties to the pentameric-repeating unit. GumF, GumG, and GumL can decorate the mannose and glucuronic acid residues with acetyl and pyruvate moieties [14]. Afterward, the lipid-linked pentasaccharide is ‘flipped’ to the periplasm by GumJ, GumE polymerizes the pentameric units, and finally, the xanthan molecule is exported through a specific channel in the outer membrane formed by GumB and GumC [14].

Xanthan synthesis is a highly energy-demanding process that must be tightly regulated, and there may be metabolic trade-offs between cell growth and xanthan production [15]. Environmental factors that influence xanthan production in *Xanthomonas campestris* have been extensively investigated. While optimal temperatures for cell growth have been reported to be between 24 °C and 27 °C, the optimal temperature for xanthan synthesis in these studies was slightly higher, between 30 °C and 33 °C [16,17]. For different strains of *Xanthomonas campestris*, the carbon or nitrogen source type and even the ratio between carbon and nitrogen in synthetic media can significantly affect xanthan production [17,18,19,20,21,22]. Processes for industrial xanthan production often involve batch cultures in which the nitrogen source is ultimately depleted before the carbon source [22,23,24,25].

Despite a growing knowledge base on the physiology of *Xanthomonas campestris* and relevant environmental factors for xanthan production, our understanding of the regulation of xanthan synthesis is far from complete. The importance of the high C/N ratio and limited nitrogen availability in industrial batch cultures gives incentives to investigate the impact of nitrogen depletion on xanthan production by *Xanthomonas campestris* in greater detail. Using omics technologies has previously provided extensive new insights into molecular mechanisms underlying the phenotypes of the laboratory strain *Xanthomonas campestris* pv. campestris B100. The unraveling of the complete genome of *Xcc* B100, along with an improved annotation using RNA sequence data [14,26], provided an opportunity for further research into proteome and metabolome under relevant conditions. For example, differences in gene expression between exponential growth and stationary phase have been investigated [27]. In a 2D-gel electrophoresis-based proteome analysis of *Xcc*, 281 proteins were identified, which could be attributed to 81 distinct metabolic pathways [28]. Additionally, an analysis of the extracellular proteins of *Xcc* was conducted using a similar approach, revealing 87 proteins, including 11 known degradative enzymes that play a role in pathogenesis [29]. Several proteome studies have focused on the interaction between *Xcc* and its host plants [30,31,32]. Recently, liquid chromatography–mass spectrometry (LC-MS)-based techniques have led to a significant improvement in the analysis of the *Xcc* proteome, with a much higher coverage of the predicted total proteome compared to 2D gel-based approaches [27,33].

In recent decades, researchers have made significant progress in understanding the molecular mechanism of xanthan production. However, there is still incomplete knowledge about how xanthan metabolism is regulated. This study aims to investigate the mechanism of xanthan synthesis on the protein level, which might improve the efficiency of strain design and industrial production processes. For this proposal, a label-free semi-quantitative LC-MS/MS approach was chosen.

## 2. Materials and Methods

### 2.1. Cultivation of Xcc

*Xanthomonas campestris* pathovar campestris B100 was grown in 2 L Erlenmeyer flasks with a culture volume of 200 mL at 30 °C in a laboratory shaker (Innova 44, New Brunswick Scientific, Edison, NJ, USA) at 180 rpm. The pre-culture was grown overnight in TY-medium (5 g L^−1^ tryptone, 3 g L^−1^ yeast extract, 0.7 g L^−1^ CaCl_2_). The pre-culture was used as an inoculum for the main culture in XMD minimal media (30 g L^−1^ glucose and 6 g L^−1^ ammonium nitrate) [34] with an initial OD600 of 0.1. The main culture was grown in four separate biological replicates.

### 2.2. Xcc Harvesting

Samples from six different time points (12 h, 24 h, 30 h, 36 h, 48 h, and 72 h after the start) were analyzed in four biological replicates (24 samples). The optical density, glucose concentration, nitrate concentration, and xanthan gum concentration were monitored. The volumetric xanthan production rate was calculated based on the xanthan concentration. To overcome issues in cell harvesting caused by increased viscosity of culture broth due to higher xanthan concentrations, samples were diluted with PBS up to a 1:9 ratio so that at least good cell pelleting could be observed. In practice, samples after dilution had an OD600 of 0.25 in a 2 mL volume. The diluted cell suspensions were centrifuged with 16,100 rcf at room temperature for 3 min (Centrifuge 5415 R, Eppendorf, Hamburg, Germany) immediately after harvesting. The supernatant was discarded, and the pellet was washed once in 1 mL PBS with protease inhibitor (cOmplete mini, EDTA free, Roche, Switzerland), according to the manufacturer’s instructions. The pellet was frozen in liquid nitrogen and stored at −80 °C until further processing.

### 2.3. Xanthan Gum, Glucose, and Nitrate Determination

The xanthan concentration of culture broth was determined using size exclusion chromatography on an HPLC instrument (Knauer Smartline Manager 5000, Knauer Smartline Pump 1000, Knauer Smartline RI Detector 2300 (Knauer, Berlin, Germany), Spark Holland Triathlon Autosampler (Spark, Emmen, The Netherlands) with a HiTrap column (GE Healthcare HiTrapTM CM FF 1 mL, Los Angeles, CA, USA) at a column temperature of 25 °C. The running buffer consisted of 50 mM NaH_2_PO_4_ and 150 mM NaCl at pH 4.6 and was degassed for 30 min by ultrasonication. An isocratic flow of 0.8 mL min^−1^ over 5 min was used. The xanthan fraction is detected as a negative peak, followed by a positive peak representing the sample’s remaining components. The area of the negative peak was used for xanthan quantification based on a 7-point custom calibration curve with xanthan concentrations between 0.02 and 0.2 [*w*/*v*]. According to the manufacturer’s instructions, NO^3−^ concentrations were determined using Spectroquant ^®^ Nitrate Cell Test kit (Merck, Darmstadt, Germany). According to the manufacturer’s instructions, glucose concentrations were determined with the SUPER GL ambulance kit (Dr. Müller Gerätebau GmbH, Freital, Germany).

### 2.4. Whole-Cell Protein Isolation and Digestion

Cell pellets were resuspended in 100 µL 100 mM ammonium bicarbonate. In total, 100 µL of 2,2,2-trifluoroethanol (TFE) and 5 µL of 200 mM dithiothreitol (DTT) were added for cell disruption. The suspensions were incubated at 60 °C for 60 min [33,35]. For alkylation, 20 µL of a 200 mM iodoacetamide (IAA) solution was added, and the samples were subsequently incubated for 90 min at room temperature in darkness. In total, 5 µL of a 200 mM DTT solution was added to stop the alkylation, followed by incubation at room temperature for another 60 min. The samples were diluted 1:10 with 50 mM ammonium bicarbonate for tryptic digestion with Trypsin Gold (Promega, Mannheim, Germany). An exact ratio of protein to trypsin was not possible to adjust due to interference from TFE during protein concentration determination. To ensure the most complete digestion possible, 10 µg trypsin was added to the protein solution and incubated overnight at 37 °C.

### 2.5. Sample Preparation for LC-MS/MS Analysis

According to the manufacturer’s instructions, digested protein samples were purified with C18 columns (Sep-PAK^®^ Vac 1cc, Waters, Elstree, UK). Peptides were quantified using a Nanodrop 2000 (peqlab, VWR, Radnor, TN, USA) instrument.

### 2.6. LC-MS/MS Analysis

The label-free LC-MS/MS analysis was performed with a Mass Spectrometer (Q Exactive™ Plus Hybrid Quadrupole-Orbitrap™, Thermo Scientific™, Walthem, MA, USA) and an upstream nano-HPLC (Ultimate TM 3000 RSLCnano System, Thermo Scientific, USA). The QExactive Plus was equipped with a Spectroglyph source (Spectroglyph, LLC., Washington, DC, USA), whereby the S-lence was replaced by an ion tunnel [36]. A 1 µg peptide was injected and separated into fractions using a 60 min acetonitrile gradient of 5–40% and a flow of 300 nL min^−1^. Two different models of the Acclaim™ PepMap™ 100 C18 (Thermo Scientific, USA) column were used as pre-column and main column. The pre-column had a diameter of 0.3 mm, a length of 5 mm, and a particle size of 5 µm. The diameter of the main column was 0.075 mm by a length of 250 mm and a particle size of 3 µm. The temperature of the column oven was set to 35 °C. Measurements were recorded in data-dependent acquisition mode. On FullMS, the resolution was 70,000, the AGC target was 3 × 10^6^, and the maximum IT was 64 ms. The dd-MS2 settings were 17,500 for the resolution, 2 × 10^5^ for the AGC target, and 100 ms for the maximum IT. TopN was set to 10.

### 2.7. Data Analysis

The database search was performed with MaxQuant [37] version 1.6.14.0. Methionine oxidation and N-terminal acetylation were set as variable modifications, and the cysteine carbamidomethylation was set as a fixed modification. For digestion, trypsin was selected with a maximum of 2 missed cleavages. Label-free quantification and fast LFQ were enabled. The normalization was performed in classic mode, with a minimal peptide ratio count of 2. Unique peptides were used for quantification. For the FDR calculation, the decoy mode was set to random. The 4413 entries containing the database from UniProt of *Xanthomonas campestris* pv. campestris B100 (Taxon identifier: 509169) were used as the reference database. The remaining settings were set to default.

The statistical analysis of the MaxQuant database search was performed in Perseus [38] 1.6.15.0 and Omics Fusion [39] using the LQF intensities. The proteins that MaxQuant identified as “Only identified by site, “Reverse”, or “Potential contamination” were filtered out for Perseus and Fusion. Proteins with fewer than two unique peptides were removed to ensure the validity of the data. The normal distribution of the data was obtained by log2-transformation. The missing values were replaced by the trimmed mean of the available values if at least 3/4 of the feature values were available. A Hochberg-corrected two-sample *t*-test was performed to define the significant difference between two samples. Only changes in abundance with a *p*-value of 0.05 and a log2 fold change of 0.5 were considered significant.

The dataset and MaxQuant analysis were uploaded to ProteomeXchanger [40] via PRIDE [41] with the dataset identifier PXD041470 (website: http://www.ebi.ac.uk/pride (accessed on 22 April 2023), project accession: PXD041470).

## 3. Results

### 3.1. Nutrient Consumption and Xanthan Production of Suspension Cultures of Xcc B100 over a Time Course

*Xcc* B100 was grown in a synthetic medium with limited nitrogen availability (6 g L^−1^ ammonium nitrate) in four biological replicates for 72 h to investigate changes in phenotype and proteome composition. Proteome sampling was performed at six points in time, and the optical density, nitrate, glucose, and xanthan levels were monitored throughout the experiment (Figure 1A). The optical density measurements revealed four distinct phases during the experiment. After inoculation, the cultures showed a 6 h lag phase followed by the first logarithmic growth phase with a specific growth rate of 0.13 h^−1^ ± 0.00062 h^−1^ between 6 and 24 h, and a second logarithmic growth phase between 24 h and 42 h with a particular growth rate of 0.05 h^−1^ ± 0.0036 h^−1^. After 42 h, the optical density remained constant, and the cultures reached a stationary phase. This observation coincided with nitrate depletion from the medium at 48 h. When the experiment was terminated after 72 h, glucose was not entirely consumed, and a residual concentration of 17.8 g L^−1^ ± 0.925 g L^−1^ was measured. The xanthan concentration increased throughout the experiment (Figure 1B), reaching 3.93 g L^−1^ ± 0.71 g L^−1^ after 72 h. The volumetric xanthan production rate increased from 0.006 g (L h)^−1^ to 0.017 g (L h)^−1^ between 24 h and 48 h but decreased again to 0.003 g (L h)^−1^ at the stationary phase.

### 3.2. The Abundance of Ribosomal Proteins Varies Per Growth Phase

Throughout the experiment, changes in the abundance of proteins were investigated through label-free LC-MS/MS measurements. Quantitative changes in protein abundance were judged from individual proteins’ relative abundance at different time points. In total, 2416 proteins were identified, representing 54.75% of the entries in the reference database (Appendix A). The cutoff for the student *t*-tests was set to a *p*-value of 0.05 and a log2 fold change of 0.5. These findings were used to investigate the protein changes in *Xcc*. The abundance of ribosomal proteins, which, according to the literature, shows a strong correlation to growth rate [42,43], was explored to benchmark the quality of the LC-MS/MS measurements. For this analysis, a set of core-conserved ribosomal proteins was selected [44,45]. Figure 2A,B show these core ribosomal subunits identified in the current *Xcc* proteome dataset. Most Rps and Rpl proteins showed the highest abundance at the start of the experiment when the growth rate was also highest. Ribosomal proteins generally decreased until the second logarithmic growth phase at 30 h. According to the student *t*-tests (Appendix A), these proteins did not change in abundance significantly during the second logarithmic phase, except for RplC, which displayed an increased abundance. The ribosomal protein abundance then decreased until the cultivation ended at 72 h. This decrease in the ribosome fraction during the decreasing growth rate is in line with other literature [42] and, therefore, supports the good quality of the dataset.

### 3.3. The Effect of Nitrate Depletion on the Abundance of Proteins Involved in Nitrogen Metabolism

Because of the vital role nitrogen availability plays in inducing the stationary phase, nitrogen metabolism was more closely investigated in this study (Figure 3). The abundance of proteins involved in nitrate metabolism increased from 12 h to 36 h. Most protein intensities then decreased again rapidly after the onset of the nitrate depletion-induced stationary phase, although the intensity of glutamine synthetase GlnA changes insignificantly between 36 h and 72 h. The glutamate dehydrogenase Gdh showed no significant changes in abundance throughout the whole cultivation.

### 3.4. Proteins Involved in the Synthesis of Sugar Nucleotide Precursors and Xanthan

During the experiment, the changes in protein levels responsible for synthesizing the sugar nucleotides UDP-glucose, UDP-glucuronate, and GDP-mannose, as well as the products of the gum cluster, were studied, with particular attention to differences between the time when nitrogen was available and nitrogen depletion. The transmembrane glucose kinase Glk, which facilitates glucose uptake, showed no significant changes in abundance (Figure 4). Statistically significant differences in protein abundances were observed for the glucose-6-phosphate isomerase Pgi with a fold change of 0.5 between 36 h and 48 h and the mannose-6-phosphate isomerase XanB from 12 h to 24 h with a fold change of 0.5. Other than these small changes in abundance, there were no noticeable differences in this pathway throughout the experiment.

Next to other pathways like lipopolysaccharide synthesis, the sugar nucleotides produced via this pathway are utilized for xanthan synthesis. Seven out of twelve gum proteins could be quantified and were analyzed to examine the response of the xanthan synthesis mechanism throughout the experiment (Figure 5). GumD, the first enzyme of the pathway, showed a significant increase in abundance during the first phase of exponential growth (12–24 h) but reached its highest intensity during the transition between the second logarithmic growth phase and the onset of the stationary phase. This was followed by a significant decrease until the termination of the experiment. GumK and GumI remained relatively unchanged, while GumH showed a small but insignificant increase after transitioning into the stationary phase. GumL, the only detected decorating enzyme, increased in abundance during the experiment’s early phase and then gradually decreased. GumB and GumC, which form the xanthan exporter pore, showed a considerable increase in abundance after transitioning into the stationary phase (36–48 h). By the end of the experiment, GumB and GumC had significantly increased in abundance. In conclusion, out of all detected gum proteins, only GumD, GumB, and GumC showed measurable responses to the nitrogen-induced stationary phase.

The results presented here reflect changes in the proteome of the widely used laboratory strain Xcc B100 over batch cultivation with restricted nitrogen and excess carbon resources based on a semi-quantitative LC-MS/MS approach. The improved methodology and increased time resolution contribute to a deeper understanding of the effect of nitrogen depletion and growth phase on xanthan synthesis at the protein level.

## 4. Discussion

Xanthan, produced by *Xanthomonas campestris*, is a widely used polysaccharide. A detailed study of the *Xanthomonas campestris* proteome during xanthan production can aid in understanding the molecular mechanisms during fermentation. The validity of proteome data is primarily determined by the coverage of the reference protein database [46]. Recent mass spectrometry-based proteome discovery studies typically obtain sequence coverages of their predicted gene models ranging from 50 to 70% [47] under defined conditions like cultivating in a minimal medium. The protein coverage of organisms closely related to *Xcc*, such as *Xanthomonas oryzae* pv. oryzae or *Xanthomonas citri*, range from 40 to 52% [48,49]. Therefore, this work’s protein coverage of 54.75% for *Xcc* B100 is in a similar range to previous studies. The quality of proteome data is influenced by factors such as sample preparation, properties of individual peptides, and the analysis method [47,50,51]. Here, the quality of the study’s dataset was inspected based on ribosomal subunits, which changed in abundance similarly to the literature described [42,52,53].

Since nitrogen availability influenced the growth rate and xanthan production, a closer look at nitrogen metabolism was taken [18,19,54]. Most proteins related to nitrogen metabolism, including NapA/Xcc-b100_2307/Xcc-b100_2308/GltD/GltB, displayed changing abundances in response to nitrate depletion after 36 h. The function of these proteins covers the import and processing of nitrate to ammonia and the conversion of glutamine to glutamate. However, the profiles of Gdh and GlnA did not follow a similar pattern. These two proteins are known regulators of the glutamate pool in enteric bacteria [55]. Due to nitrogen limitation, the gene *gdh* is repressed while *glnA* is activated [55,56]. Based on the present experimental conditions, a change in the abundance of the proteins Gdh and GlnA was not observed. Finding an explanation for the difference between the gene expression known from the literature and the abundance profile of GlnA and Gdh is impossible with the available proteome data. Speculatively, if Gdh and GlnA play a similar role in *Xcc* as in enteric bacteria, their action may be regulated by post-translational modifications rather than abundance. In vitro phosphorylation has been demonstrated for GlnA from *Bacillus subitilis* [57], and *Klebsiella aerogenes* GlnA function is regulated by adenylation and inhibits the transcription of *glnA* [58].

Apart from nitrogen metabolism, this study also focused on the process of converting glucose into sugar nucleotides on the proteome level. Under the tested conditions, the abundance of the sugar nucleotide-synthesizing proteins remained unchanged regardless of the growth phase. A similar observation was made in earlier studies for the transcription rates of the corresponding encoding genes in *Xcc*. Most transcripts for genes associated with sugar nucleotide synthesis were unaffected or lower in the stationary phase than in the exponential phase [27]. Post-translational modifications might play a role in the activity of this pathway instead. Previous research has shown that the phosphorylation state of XanA varies depending on the culture’s growth phase, which could affect its activity [59]. Although it cannot be excluded that such other factors play a role, the data presented here do not support any regulation of the activity of this pathway through protein abundance in response to nitrogen limitation.

This study presents a more detailed observation of the abundance of gum proteins over time during the cultivation of *Xcc*. It thereby provides information on the molecular mechanisms of xanthan synthesis on the proteome level. As was observed for proteins involved in sugar nucleotide synthesis, most gum proteins showed no significant changes in abundance throughout the experiment. As one of the few exceptions, the abundance of GumD, the first glucosylphosphate transferase that loads the C55-lipid carrier, peaked in the early stationary phase and decreased again afterward. Because the peak in abundance was present in all replicates, and since such a ‘jump’ at this time point was not observed for other proteins, it is unlikely that this is due to an artifact. This contrasts with a previous study where no such ‘jump’ was observed for GumD [33]. Perhaps in this study, the more comprehensive time resolution allowed for the capture of the dynamic behavior of GumD. It has been hypothesized that an essential function of GumD is to control the metabolic flux into xanthan synthesis in *Xcc* [60]. If the increase in GumD levels is triggered by nitrogen limitation or the onset of the stationary phase, this could reflect a regulated metabolic ‘switch’ to polysaccharide production under these conditions. Other factors, for example, oxygen limitation, may have further limited xanthan production in *Xcc* in this experiment and prevented a further metabolic response. However, this remains speculation until the impact of the abundance of GumD on polysaccharide synthesis is more closely investigated.

The abundance of the export proteins GumB and GumC also increased significantly toward the end of the experiment. A relationship between a higher expression of GumB and GumC and a higher xanthan viscosity has previously been reported. It was suggested that the increased abundance of both pore-forming proteins is the origin of higher-molecular-weight xanthan [61]. In this experiment, we may have captured a change in the quality of xanthan triggered upon nitrogen depletion rather than quantity.

When comparing the changes in abundance of the gum proteins detected in this study with previously reported transcription start sites (TSSs), the patterns align with the gum cluster’s proposed genetic organization. For instance, *gumB* and *gumC* are controlled by the same TSS, and their products’ abundances change similarly over time [62]. Likewise, the TSS in front of *gumH* also controls *gumI*, *J*, *K*, *L*, and *M* transcription, and the detected and quantifiable gene products of these genes also similarly change in abundance. For the gum proteins detected in this study, protein abundances corresponded to the proposed operon model. This was not the case for the transcript rates presented in an earlier study [27]. The protein abundances and transcript rates of *gumD*, *gumC*, *gumB*, and *gumL* appear to be similar, while *gumH*, *gumI*, and *gumK* show no changes in protein levels but a decrease in the transcript rate. While the transcription rate is decreasing, and the protein abundance remains the same, this might hint that *Xcc* changes its metabolism until new nutrients are available. This could be a strategy to compare the energy demand of the xanthan synthesis process [15]. The interpretation of transcript and proteome data may differ due to the more significant number of data points in the proteome study. Therefore, increasing the data points for transcript rates could lead to similar observations.

As for the sugar nucleotide synthesis, post-translational modifications could also be considered for the regulation of gum proteins, and so the regulation of xanthan synthesis. However, in contrast to the sugar nucleotide synthesis proteins, there are no known phosphosites for the gum proteins, which provides a starting point for further research.

## 5. Conclusions

Xanthan is a bacterial polysaccharide and is widespread in industrial applications. Although *Xcc* is often used for xanthan fermentation, little is known about xanthan synthesis at the protein level. This study’s extensive profiling of *Xcc’s* proteome over time by label-free LC-MS/MS offers valuable insights into the regulation and mechanism of xanthan synthesis. The available data could, therefore, support the assumption that changes in the protein abundance of GumB/C over the cultivation time correlate with a change in product quality. Further, the data show that the protein abundance in metabolic pathways leading up to xanthan synthesis may not be a bottleneck for productivity. It, therefore, provides an incentive to search for alternative points of regulation in the production of xanthan. Other proteoforms of the proteins in the xanthan synthesis pathway might play an essential role in regulating xanthan synthesis, like protein phosphorylation. Further investigations into this topic could be covered by a phosphoproteome study. Also, a metabolomic study with a comparable number of data points, like in the presented study, could provide new insights into the xanthan synthesis mechanism.

## Figures and Tables

**Figure 1 proteomes-12-00012-f001:**
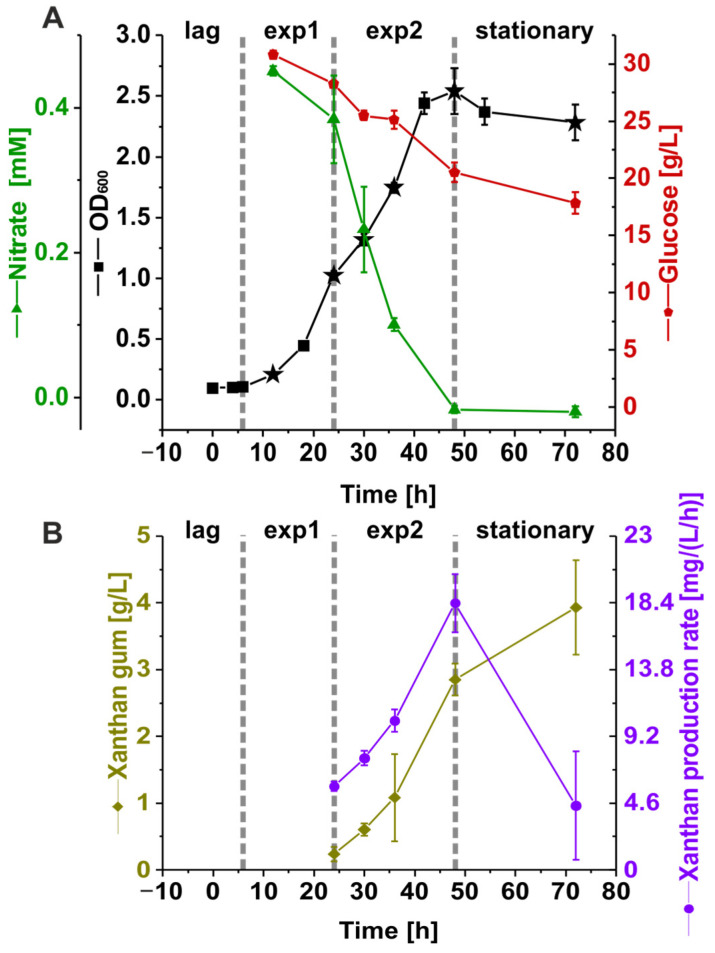
(**A**) *Xanthomonas campestris* pv. campestris B100 was grown in four biological replicates on a synthetic medium. The cultures were monitored for 72 h, with gray dotted lines marking the lag, exponential growth (exp1, exp2), and stationary phases. Proteome samples were taken at times indicated by star symbols. In addition, the nitrate (mM) and glucose concentrations (g/L) were measured. (**B**) The xanthan concentration (g/L) was monitored from 24 h after inoculation until the end of the experiment. Based on these values, a xanthan production rate in g/(OD600/h) was estimated. The error bars of A and B indicate the standard deviation of the biological replicates. lag = lag phase; exp1 = first exponential phase; exp2 = second exponential phase; stationary = stationary phase.

**Figure 2 proteomes-12-00012-f002:**
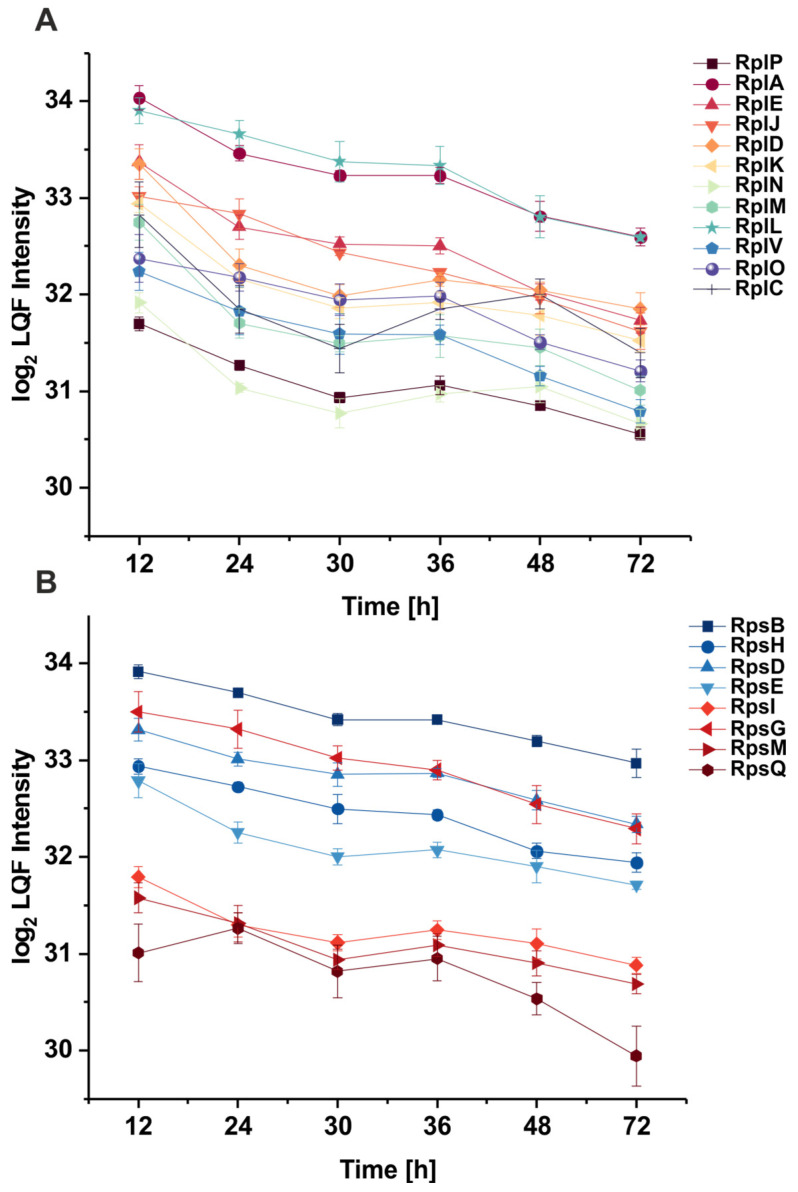
Abundance of the ribosomal proteins of the large (**A**) and small (**B**) subunit of *Xanthomonas campestris* pv. campestris B100 over time during cultivation in a minimal synthetic medium. The displayed proteins were selected based on matches to universally conserved ribosomal core proteins [45].

**Figure 3 proteomes-12-00012-f003:**
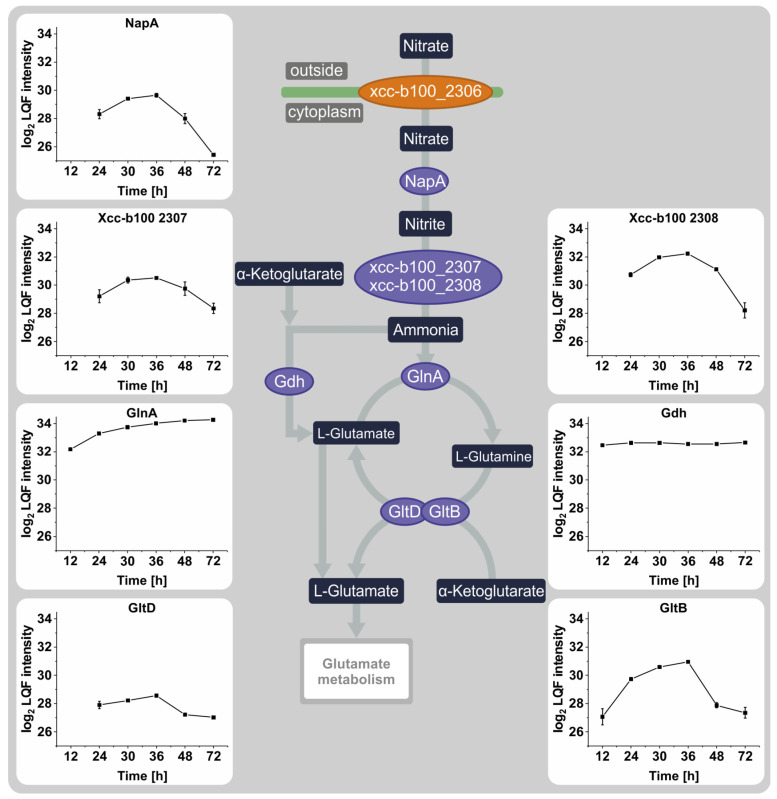
Representation of nitrogen metabolism in *Xanthomonas campestris* pv. campestris B100 (KEGG: xca00910). The profile plots display the log2-transformed LFQ intensities of four biological replicates of the respective proteins. The error bars indicate the standard deviation of the biological replicates. Proteins marked in orange were not detected in this experiment.

**Figure 4 proteomes-12-00012-f004:**
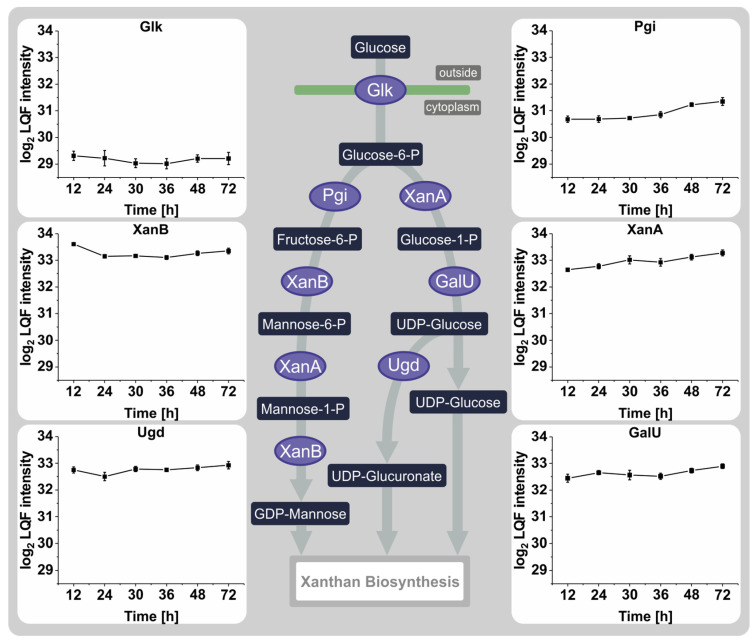
Representation of sugar nucleotide synthesis in *Xanthomonas campestris* pv. campestris B100. The key steps, from glucose uptake to synthesizing the individual sugar nucleotide precursors, are indicated. The profile plots display the log2-transformed LFQ intensities of four biological replicates of the respective proteins. The error bars indicate the standard deviation of the biological replicates.

**Figure 5 proteomes-12-00012-f005:**
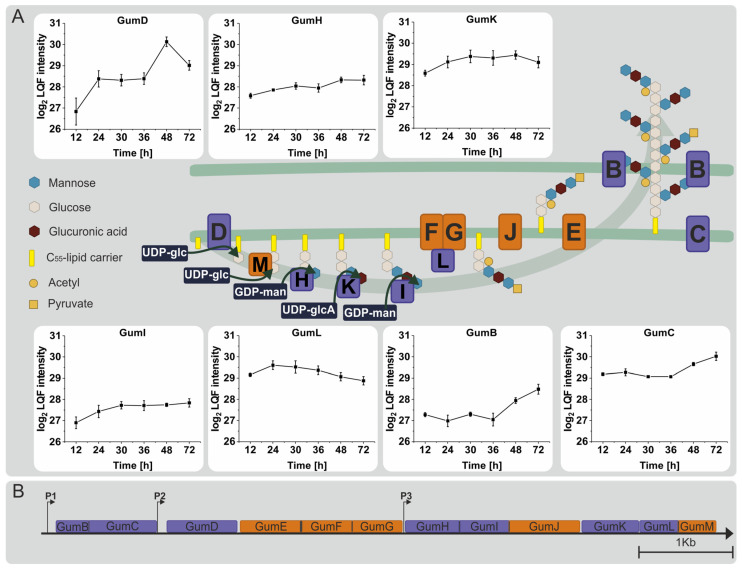
(**A**) Representation of xanthan synthesis in *Xanthomonas campestris* pv. campestris B100 according to Vorhölter et al. (2008) [14]. The profile plots display the log2-transformed LFQ intensities of four biological replicates of the respective proteins. Proteins marked in orange could not be detected or quantified, while proteins marked in purple color were detected in this experiment. UDP-glc = UDP-glucose, UDP-glcA = UDP-glucuronate, GDP-man = GDP-mannose. (**B**) Representation of the gum gene cluster modified according to Alkhateeb et al. (2017) [26]. P1, P2, and P3 represent the positions of the promoter start.

## Data Availability

Publicly available datasets were analyzed in this study. These data can be found here: [http://www.ebi.ac.uk/pride/PXD041470] (accessed on 22 April 2023).

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
