# Peer review of "Comprehensive Proteome Profiling of a Xanthomonas campestris pv. Campestris B100 Culture Grown in Minimal Medium with a Specific Focus on Nutrient Consumption and Xanthan Biosynthesis"

_proteomes, 2024, doi:10.3390/proteomes12020012_

Round 1
Reviewer 1 Report
Comments and Suggestions for Authors
Overall, the paper presents a well-conducted study that advances our understanding of xanthan synthesis in Xcc, and the findings of the study are well-supported.
Only several small issues can be addressed to improve the quality of the paper:
1) in the method part, step 2.4, for whole cell protein digestion, the author used "cell pellets were resuspended in ..." while the "size" of cell pellet is not clear, although, in step 2.2, the author mentioned, cell solution had an OD600 of 0.25, but didn't mention the volume to make cell pellet. Second thing in this part, the author mentions "tryptic digestion" was using "manufacturer’s instructions", usually, for this step, there is a trypsin to total protein ratio of 1:20, but how is the total protein in the cell pellet determined? this part needs to be clarified.
2) Line 145-147, here are some gramma issues "An ... C18 column was used as pre-column and main column" looks like there are two different-sized columns. "The diameters of the main column were" should be "The diameter of the main column is..."
Author Response
Review 1: in the method part, step 2.4, for whole cell protein digestion, the author used "cell pellets were resuspended in ..." while the "size" of cell pellet is not clear, although,
Author: An exact specification of the pellet size was not possible, but based on the specification that the diluted culture had an OD 600 of 0.25 in 2 ml provides an idea of the pellet dimension.
Review 1:in step 2.2, the author mentioned, cell solution had an OD600 of 0.25, but didn't mention the volume to make cell pellet.
Author: In line 104 the phrase " in a 2 ml volume" was added.
Review 1: Second thing in this part, the author mentions "tryptic digestion" was using "manufacturer’s instructions", usually, for this step, there is a trypsin to total protein ratio of 1:20, but how is the total protein in the cell pellet determined? this part needs to be clarified.
Author: Due to interferences of the TFE during protein concentration determination, an excess of trypsin was estimated. Also, overnight incubation was used to ensure that digestion was as complete as possible (lines 133-136).
Review 1:Line 145-147, here are some gramma issues "An ... C18 column was used as pre-column and main column" looks like there are two different-sized columns.
Author: The sentence was restructured to clarify that the pre-and main columns are made of the same column material but differ in models (lines 149-150)
Review 1:"The diameters of the main column were" should be "The diameter of the main column is..."
Author: Improved.
Reviewer 2 Report
Comments and Suggestions for Authors
In this study, the authors utilized the label-free LC-MS technique to comprehensively investigate the proteome change of Xanthomonas campestris, which is the microbiome frequently used for xanthan fermentation, over the course of time. The detailed study and explicit discussion made the study interesting and intriguing, and the article is well-written in English and easy to follow. Although the protein abundance in metabolic pathways seems not to be a bottleneck for xanthan productivity, it provides insights into the process of xanthan synthesis on the proteome level. Neat work! and I recommend it to be published as it is or subject to minor revisions if other reviewers have any.
Author Response
Thank you very much for your kind review! As pointed out in your review, that the gum-proteins seem not to be the bootle neck for Xanthan production under fermentation conditions, is more emphasised now.
Reviewer 3 Report
Comments and Suggestions for Authors
Xanthan is a complex carbohydrate, a polysaccharide with a high molecular weight obtained through bacterial fermentation of natural strains of Xanthomonas campestris, from which its name derives. Xanthan has numerous applications in the food field, where it is used as a thickening and stabilizing additive, but also in the pharmaceutical and cosmetic fields.
In this work, a little-researched aspect such as the xanthan synthesis process at the proteome level was studied. Therefore, a label-free LC-MS/MS method was employed to study protein changes during xanthan fermentation in minimal medium.
The topic covered certainly falls within the aims and scope of the Journal, furthermore it provides a series of interesting data that deserve further speculation.
Overall, the experimental work is well done and the results allow to enrich knowledge about this compound. However, some changes are required as below.
Lines 80-80: This paragraph should be moved to the end of the results section and a short paragraph should be added in its place to summarize the purpose of the study.
In the conclusions section, the authors should expand the sentence regarding future prospects and the aspects to be investigated based on the data obtained.
Comments on the Quality of English LanguageThe English used is easily readable.
Author Response
Review 3:Lines 80-80: This paragraph should be moved to the end of the results section and a short paragraph should be added in its place to summarize the purpose of the study.
Author: The paragraph was moved to lines 287-291, and a short paragraph explaining the study's purpose was added from lines 80 to 85.
Review 3: In the conclusions section, the authors should expand the sentence regarding future prospects and the aspects to be investigated based on the data obtained.
Author: The conclusion was expanded to give more detail about further possible studies, like a phosphoproteome study or a metabolome study (lines 384-393).
Reviewer 4 Report
Comments and Suggestions for Authors
In this manuscript, the Authors present a detailed proteomics survey of protein changes in Xanthomonas campestris during xanthan production in culture. The Authors should be commended for exceptionally high-quality proteomics analyses paired with readily interpretable figures (e.g. Fig. 3-5, where alterations to the proteome are presented along with the appropriate schematics of nitrogen metabolism and xanthan biosynthesis). On the other hand, the paper is essentially a 'negative data' paper demonstrating that, in the words of the Authors, "a potential bottleneck for xanthan productivity does not reside in the abundance 23 of proteins directly involved in the synthesis pathways". However, the quality of the data presented as well as the level of interest in xanthan fermentation and the importance of xanthan gum production and optimization for industry still merit the publication of this paper in this Reviewer's opinion. In short, the 'negative data' are still meaningful data and this Reviewer is of the opinion that the study will be of interest to readers of this journal.
Author Response
Thank you very much for your kind review!